# Arterial Stiffness and HbA1c: Association Mediated by Insulin Resistance in Hispanic Adults

**DOI:** 10.3390/ijerph191711017

**Published:** 2022-09-03

**Authors:** Alexandro J. Martagón, Carlos A. Fermín-Martínez, Neftali Eduardo Antonio-Villa, Roopa Mehta, Paloma Almeda-Valdés, Arsenio Vargas-Vázquez, Liliana Muñoz-Hernández, Donají V. Gómez-Velasco, Daniel Elías-López, Gabriela A. Galán-Ramírez, Fabiola Mabel del Razo-Olvera, Ivette Cruz-Bautista, Rogelio González-Arellanes, Carlos A. Aguilar-Salinas

**Affiliations:** 1Unidad de Investigación de Enfermedades Metabólicas, Instituto Nacional de Ciencias Médicas y Nutrición Salvador Zubirán, Mexico City 14080, Mexico; 2Escuela de Medicina y Ciencias de la Salud, Tecnologico de Monterrey, Mexico City 64700, Mexico; 3Facultad de Medicina, Universidad Nacional Autónoma de México, Mexico City 04100, Mexico; 4Departamento de Endocrinología y Metabolismo, Instituto Nacional de Ciencias Médicas y Nutrición Salvador Zubirán, Mexico City 14080, Mexico

**Keywords:** arterial stiffness, HbA1c, insulin resistance, vascular health

## Abstract

Arterial stiffness may be associated with glucose metabolism parameters, such as HbA1c, mainly via insulin resistance. We aimed to investigate the association between arterial stiffness and HbA1c and explore the mediator effect of insulin resistance. In this cross-sectional study, arterial stiffness (pulse-wave velocity; PWV), HbA1c, and insulin resistance (METS-IR) were determined in Hispanic adults. In addition to sex and age, various biochemical measurements (glucose, lipid profile, etc.) and adipose tissue (fat mass and visceral fat mass) were considered as potential confounding variables. A multivariate regression analysis shows that HbA1c is associated with PWV, even after adjusting for several confounding variables. Importantly, the results show that insulin resistance mediated 17.9% of the effect of HbA1c over PWV. In conclusion, HbA1c may be a potential resource for predicting arterial stiffness due to the influence of insulin resistance in Hispanic subjects.

## 1. Introduction

Arterial stiffness is a risk factor for cardiovascular events and all-cause mortality [1]. To evaluate this phenomenon, the pulse-wave velocity (PWV) has been proposed as a reproducible, validated, and non-invasive method based on a regional functional assessment to quantify arterial stiffness over a certain arterial length, usually the carotid–femoral distance [2]. Consequently, it has been reported that carotid–femoral PWV could be influenced by age, sex, and ethnicity [3,4]. Furthermore, several cardiovascular characteristics, such as systolic blood pressure, lipid profile, and central obesity are strong predictors of PWV values [5,6]. In the case of ethnicity, Caucasian subjects had greater PWV values in comparison with African Americans (7.3 IC: 6.9–7.6 vs. 6.7 IC: 6.5–7.0, *p* < 0.05) [7], and African American adults had greater PWV values than Hispanic adults (7.1 ± 1.1 vs. 6.2 ± 0.7, *p* < 0.05) [8]. Similarly, conducted epidemiological studies have consistently associated aortic stiffness and vascular health with glucose metabolism measurements, including fasting plasma glucose (FPG) and glycosylated hemoglobin A (HbA1c), both in healthy and diabetic Caucasian, Asian, and African-American adults [9,10,11,12]. However, some authors propose that in subjects with diabetes, glycemic control has minimal to no effect on arterial stiffness [13,14]. These findings suggested that their mechanism is determined mainly by insulin resistance due to endothelial dysfunction, an abnormal insulin vasodilating effect mediated by endothelium-derived nitric oxide [15,16]. It has been reported from cohort and transversal studies that HbA1c is associated and correlated with arterial stiffness assessed by PWV [17,18,19]. However, according to our research, these studies do not explore the mediator effect of insulin resistance (IR) on the association between HbA1c and PWV. In addition, it is important to explore if this association is maintained in other ethnic groups, such as Hispanic adults. Therefore, in this study, we sought to assess the association between arterial stiffness and HbA1c and explore the mediator effect of insulin resistance in Hispanic adults. 

## 2. Materials and Methods

For this cross-sectional study, we used information from an open-population cohort consisting of consecutive recruited subjects between January 2018 and January 2020 at Unidad de Investigación de Enfermedades Metabólicas of the Instituto Nacional de Ciencias Médicas y Nutrición Salvador Zubirán (INCMNSZ) in Mexico City (Figure 1). Subjects with metabolic comorbidities such as type 2 diabetes (T2D) (previous medical history or HbA1c ≥ 6.5%), prediabetes (previous medical history or HbA1c ≥ 5.7%), both primary and secondary dyslipidemias, and different body mass indexes (BMI) were included. Medical and family history; sociodemographic, diet, physical activity data; and all measurements (described above) were obtained by trained staff. The Human Research Ethics Committee of the INCMNSZ approved the study with reference number 361, and all participants provided written informed consent.

All measurements were assessed on the same day for each patient with an 8–12 h of fasting condition, including ≤48 h of non-consumption of caffeinated beverages and refraining from smoking.

### 2.1. Dependent Variable: Arterial Stiffness

Arterial stiffness was determined by PWV, and all participants were supine positioned for 10 min. Baseline supine brachial artery blood pressure (BP) and heart rate (HR) were measured using a SphygmoCor XCEL, AtCor Medical Pty Ltd., Naperville, IL, USA. PWV measurements were taken after two consecutive readings, 5 min apart, presenting systolic BP (SBP) of ±9 mmHg, diastolic BP (DBP) ±6 mmHg, and HR ±8 bpm. Mean arterial blood pressure (MAP) was calculated by adding two thirds of DBP to one third of SBP. For the assessment of PWV, carotid pulse waves were obtained through applanation tonometry, while femoral pulse waves were simultaneously obtained by an automated partially inflated cuff over the femoral artery at the leg midway between the hip and knee. PWV was calculated by obtaining the ratio of the corrected distance between both pulse measuring sites to time the delay between the carotid and femoral pulse waves. The distance was measured with a non-stretchable measuring tape, (1) from the carotid site to the suprasternal notch, (2) from the femoral artery at the inguinal ligament to the proximal edge of the thigh cuff, and (3) from the suprasternal notch to the proximal edge of the thigh cuff. A subtraction of distances 1 and 2 from distance 3 was used for the calculation of PWV. 

The augmentation index (AIX) was also used in this study. This parameter is derived from the augmentation pressure, and it estimates the wave reflection from the periphery as a surrogate indicator of left ventricular systolic loading; an earlier return of the reflected wave occurs with increased PWV. Normalized values of AIX for heart rate at 75 bpm (AIX75) were used to reduce the effect attributable to HR [20]. Finally, we also used the Buckberg index, also known as the subendocardial viability ratio (SEVR), a measure derived from the systolic and diastolic pressure–time curves; lower values of this metric reflect an imbalance between myocardial oxygen supply and demand. 

### 2.2. Independent Variable: HbA1c

Glycated hemoglobin A levels were measured in sample blood using high-performance liquid chromatography (HPLC) (Variant II Turbo, BIORAD).

### 2.3. Mediator and Potential Confounding Variables

Insulin resistance was considered as a mediator of the association between arterial stiffness and HbA1c. IR was determined by the metabolic score for insulin resistance (METS-IR), this score was calculated according to the following equation: METS-IR = (Ln ((2 × FG) + TG) × BMI)/(Ln(HDL-C)), where FG is fasting glucose and TG represents fasting triglyceride concentrations [21]. 

The potential confounding variables total cholesterol, triglycerides, and HDL-C were measured using colorimetric assays (Unicel DxC 600 Synchron Clinical System Beckman Coulter), and LDL-C was calculated using Martin’s equation [22]. The plasma glucose concentration was measured using an automated glucose analyzer (Yellow Springs Instruments, Yellow Springs, OH, USA), and serum insulin was determined using a chemiluminescent immunoassay (Beckman Coulter Access 2). 

In addition, all subjects were weighed on calibrated scales using an SECA mBCA 514 medical body composition analyzer, and height was determined with a floor scale SECA stadiometer. Waist circumference (WC) was evaluated using a non-stretchable SECA 201 measuring tape with 0.1 cm precision; we placed the tape directly over the skin at the mid-point between the ribcage and the iliac crest with the tape parallel to the floor. The BMI was calculated as the weight in kg divided by the squared product of height in meters. The waist-to-height ratio (WtHt) was calculated using the WC divided by height, both in centimeters. Furthermore, dual-energy X-ray absorptiometry (DXA) (GE Healthcare, Boston, MA, USA), enCORE software version 16 (GE Medical Systems Ultrasound & Primary Care Diagnostics, LLC, Wauwatosa, WI, USA) was used to measure visceral adipose tissue (VAT), and it was divided by squared height in meters to obtain a normalized index for visceral fat, as it is a better predictor of metabolic dysfunction than uncorrected VAT values. We refer to this normalized predictor as a visceral fat-mass index (VFMI).

### 2.4. Statistical Analysis

#### 2.4.1. Descriptive Analysis

Descriptive statistics were presented as absolute frequencies and percentages for categorical variables and as means ± SD or medians (interquartile range; IQR) for continuous variables. Student’s *t*-test and the Mann–Whitney U test were conducted to compare continuous variables, where appropriate. The chi-square test was used to compare the categorical variables. Missing data were addressed by using multiple imputation with chained equations under the assumption of data missing completely at random, implemented with the mice R package, creating five multiply imputed datasets for a maximum of five iterations; all analyses were performed with multiply imputed datasets and combined using Rubin’s rules. Variables were standardized and normalized using the ordered quantile (ORQ) normalization with the BestNormalize R package. Statistical significance was set at a two-tailed *p*-value < 0.05; all statistical analyses were conducted using R statistical software version 4.0.2 (R Foundation for Statistical Computing, Vienna, Austria). 

#### 2.4.2. Association Assessment

Pearson coefficients were calculated to obtain correlation matrices including the variables of interest. We evaluated HbA1c as a predictor of PWV, AIX75, and SEVR using mixed-effects regression analyses, where we included the different clinical phenotypes contained in the study as a random effect to address the heterogeneity in our sample. In these regression models, we assessed non-linear relationships between the HbA1c and pulse-waveform analysis measurements, and the best adjustment was chosen by the minimization of the Bayesian information criterion (BIC). These relationships were assessed as univariate models (model 1) and adjusting for age, sex, HR, and MAP (model 2). We further adjusted for VFMI and METS-IR as a clinical surrogate for insulin resistance (model 3). Diagnostics for the regression models were conducted by examining the model residuals and BIC.

#### 2.4.3. Mediation Analysis

As IR has been previously described to be associated with vascular health, we explored if IR (using METS-IR) mediates the effect of HbA1c on PWV and MAP. Therefore, we carried out model-based causal mediation analyses. All models were adjusted for age, sex, and total cholesterol as well as for PWV and MAP, where appropriate. We conducted all mediation analyses using the mediation R package to estimate average direct effects (ADE), average causal mediation effects (ACME), the total effect, and the proportion of effect mediated in all models. We estimated quasi-Bayesian 95% confidence intervals using heteroskedasticity-consistent standard errors with 1000 simulations. 

#### 2.4.4. Modifiers of the Effect of HbA1c on PWV

To extensively characterize the relationship between HbA1c and PWV, we searched for potential modifiers using interaction effects in linear regression models. Among potential modifiers, we included sex, age ≥ 50 years, obesity (BMI ≥ 30 kg/m^2^), visceral obesity (VAT ≥ 1 kg), triglycerides ≥ 150 mg/dL, total cholesterol ≥ 200 mg/dL, HDL-C < 50 mg/dL, LDL-C ≥ 100 mg/dL, non-HDL-C ≥ 130 mg/dL, cholesterol remnants ≥ 30 mg/dL, FPG ≥ 126 mg/dL, and arterial hypertension (SPB ≥ 140 mmHg or DBP ≥ 90 mmHg).

## 3. Results

### 3.1. Descriptive Analysis

The data included 840 overweight and obese subjects with a median age of 51 (IQR: 41–59), amongst whom 564 (64.8%) were female. Of these subjects, only 161 (19.2%) patients were not diagnosed with T2D or prediabetes, 259 (30.8%) patients were diagnosed with prediabetes, and 420 (50%) were living with T2D (Table 1). The subjects had a median HbA1c of 6.1% (5.7–7.6%) and a median FPG of 102.5 mg/dL (93–127 mg/dL) as well as a median PWV of 6.4 m/s (5.6–7.3 m/s), a median AIX75 of 34% (25–43%), and a median SEVR of 148 (132–165), and these measurements had significant differences across diabetes status groups (*p* < 0.05, Table 1, Figure 2A–C). The number of subjects with complete information for each variable is shown in Table 1, and details about missing data and imputed variables can be found in Figure A1.

### 3.2. Association Assessment

Significant linear correlation of HbA1c with PWV (r = 0.28, 95% CI: 0.21 to 0.34) and with SEVR (r = −0.16, 95% CI: −0.22 to −0.09) were observed. However, this was not the case for AIX75 (r = 0.05, 95% CI: −0.2 to 0.11) (Figure A2). When assessing the non-linear relationships between HbA1c and the vascular parameters, we found that a cubic adjustment for PWV, a quadratic adjustment for AIX75, and a linear adjustment for SEVR yielded the lowest BICs (Figure 2D–F). These results were reproduced when evaluating these relationships with fasting plasma glucose (Figure A3). In the mixed-effects regression models, we found that the association between HbA1c and PWV persisted, even after adjusting for sex, age, MAP, METS-IR, and VFMI (model 3, β = 0.077, *p* = 0.009). However, the effect of HbA1c on SEVR and AIX75 declined after these adjustments (Table 2). Given that the cubic, but not the linear, relationship between HbA1c and PWV declined after adjusting for covariates, we present the regression analysis for PWV as a linear relationship.

### 3.3. Mediation Analysis

Table 3 shows that METS-IR mediated 17.9% (4.0–66.0%) of the effect of HbA1c on PWV (*p* = 0.012). These results strongly suggest that the mechanism by which glucose metabolism affects vascular health leans in the direction of insulin resistance as a mediator of HbA1c. On the other hand, METS-IR was not mediated by the effect of HbA1c on MAP (*p* = 0.49). 

### 3.4. Modifiers of the Effect of HbA1c on PWV

Amongst the potential modifiers of the effect of HbA1c on PWV, we found that female subjects (β_INT_ = 0.139, 95% CI: 0.007 to 0.270), patients with FGP < 126 mg/dL (β_INT_ = 0.304, 95% CI: 0.063 to 0.544), without hypertension (β_INT_ = 0.281, 95% CI: 0.021 to 0.540), with VAT < 1kg (β_INT_ = 0.154, 95% CI: 0.016 to 0.291), with non-HDL-C < 130 mg/dL (β_INT_ = 0.176, 95% CI: 0.028 to 0.324), and with LDL-C < 100 mg/dL (β_INT_ = 0.181, 95% CI: 0.037 to 0.327) had a steeper increase in PWV with increasing HbA1c values (Figure 3). Notably, we found no significant interaction with age ≥ 50 years, obesity (BMI ≥ 30 kg/m^2^), insulin resistance (METS-IR ≥ 50), triglycerides ≥ 150 mg/dL, total-cholesterol ≥ 200 mg/dL, or cholesterol remnants ≥ 30 mg/dL (Figure A4).

## 4. Discussion

This study shows a population-based analysis composed primarily of Hispanic subjects living with prediabetes and type 2 diabetes and, to a lesser extent, healthy individuals. The main findings in the present study are (1) the contribution to the strength and consistency of the evidence on the association between glucose metabolism and arterial stiffness in Hispanic adults, (2) showing that the association between HbA1c and PWV is mediated by IR, (3) identifying the main modifiers of PWV, and (4) showing how HbA1c is valuable as an arterial stiffness predictor for healthy and prediabetic subjects, but its value is limited in subjects diagnosed with T2D. The results confirm a non-linear association of HbA1c with PWV and AIX75 and a linear relationship with Buckberg SEVR. We observed an increase of PWV and AIX75 as subjects in the subgroups progressed from healthy to prediabetes and T2D; inversely Buckberg SEVR decreased as subjects progressed into the disease, showing reduced vascular health in terms of arterial stiffness. Interestingly, PWV increased with greater HbA1c levels in healthy and prediabetic subjects, with a more pronounced growth in prediabetic patients; however, this effect seemed to reach a plateau in individuals with T2D. Likewise, AIX75 values initially displayed an increment along with HbA1c but then reached a peak and started decreasing. In this study, we also identified the main modifiers that intervene in the relationship between HbA1c and PWV. We found that PWV had a more pronounced increase in females, subjects with hypertension, with glucose > 126 mg/dL, VAT > 1 kg, LDL-C > 100 mg/dL, and non-HDL-C > 130 mg/dL.

These observations correspond with previous findings in the Whitehall II Study, in which HbA1c was associated with the progression of aortic stiffness during a 4-year follow-up, while they found other cardiovascular risk factors to be independently associated [23]. The association between HbA1c as an indicator of arterial stiffness, and therefore atherosclerotic complications of hyperglycemia, in non-diabetic subjects has been proposed in populations from Brazil and Spain [24,25]. Moreover, in The Cardiometabolic Risk in Chinese Study and in a Korean study, they concluded that HbA1c was related to PWV independent of conventional cardiovascular risk factors such as age and blood pressure [16,26]. The findings in the present study support these observations but also suggest that HbA1c is a positive predictor of arterial stiffness in subjects who do not present comorbidities or cardiovascular risk factors. However, it has been reported that HbA1c may be able to predict the degree of arterial stiffness in patients with resistant hypertension and uncontrolled diabetes [15]. Conversely, a recent study by Nuamchit T, et al. stated that, although the coexistence of T2D and hypertension is associated with arterial stiffness, HbA1c does not correlate with vascular parameters [14].

To further analyze the interrelation between these metrics, a mediation analysis allowed us to explore the relationship between vascular health (assessed by PWV, MAP, and Buckberg SEVR), METS-IR as a surrogate of insulin resistance, and HbA1c. In these analyses, we were able to determine that HbA1c exerts an effect over PWV through insulin resistance. A causal association between arterial stiffness and MAP was previously proposed, in which hypertension could develop secondary to continuously increased PWV [27], and our observations allowed us to elucidate the mechanisms underlying this relationship. Here, we hypothesize that elevations in HbA1c could trigger a complex network of pathophysiological events, where insulin resistance likely plays a fundamental mediating role that culminates in decreased vascular health.

As with every study, our study has its strengths and limitations. On one hand, we have used a validated surrogate, METS-IR, to evaluate insulin resistance [21] as well as the most validated non-invasive method to quantify arterial stiffness [2]. On the other hand, it is important to acknowledge that our results are restricted to the Mexican population and may present ethnic variabilities compared to other populations; in spite of this, it is important to characterize this population. Second, our PWV measurements were only measured in a cross-sectional methodology. It would have been desirable to evaluate the medication history for confounders since treatments for T2D and hypertension may influence risk factors; unfortunately, these data were not collected. Finally, it has been reported that short-term glycemic control (12 weeks) does not influence arterial stiffness [12]. Therefore, longer-term glycemic control studies are required to assess if HbA1c can show a positive effect in the reduction of arterial stiffness parameters.

## 5. Conclusions

HbA1c is an appropriate and valuable resource for predicting arterial stiffness in healthy and prediabetic subjects, while its utility once the patient is diagnosed with T2D reaches a limit and becomes questionable. The association between HbA1c and PWV is mediated by the insulin resistance effect. This study also highlights the importance of additional metabolic abnormalities in vascular health, supporting the implementation of integral management and not glycemic control alone, especially in prediabetic and diabetic patients.

## Figures and Tables

**Figure 1 ijerph-19-11017-f001:**
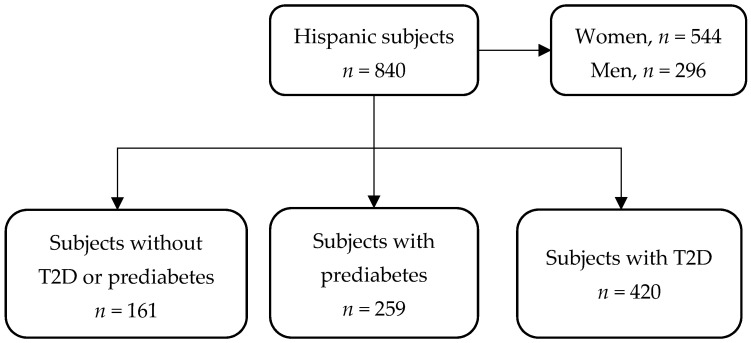
Study population flow-chart.

**Figure 2 ijerph-19-11017-f002:**
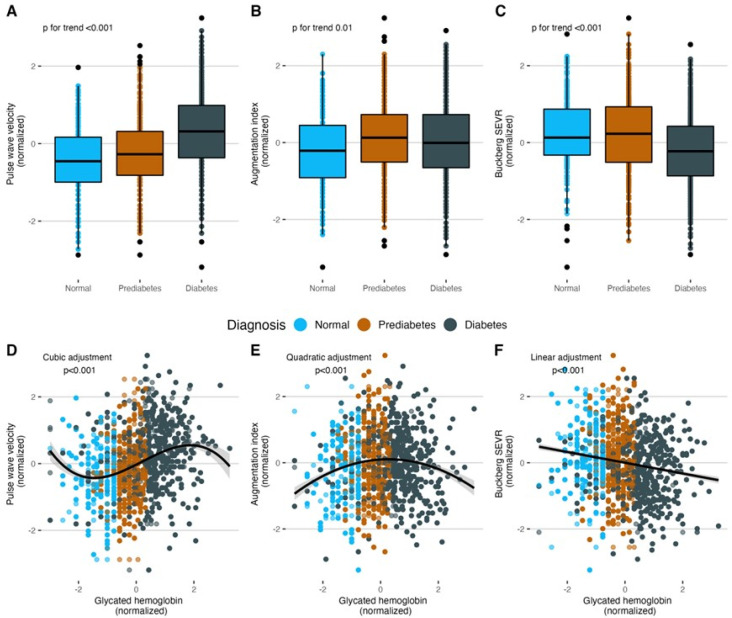
(**A**–**C**) Diabetes status has a positive association with PWV and AIX75 and a negative association with Buckberg SEVR. (**D**–**F**) According to BIC minimization, the best relationships of HbA1c are cubic for PWV, quadratic for AIX75, and linear for Buckberg SEVR. Given that the cubic relationship between HbA1c and PWV declined after adjusting for covariates, we present the regression analysis for PWV as a linear relationship. Abbreviations: PWV: Pulse-wave velocity. AIX75: Heart-rate-corrected augmentation index at 75 bpm. SEVR: Subendocardial viability ratio. BIC: Bayesian information criterion.

**Figure 3 ijerph-19-11017-f003:**
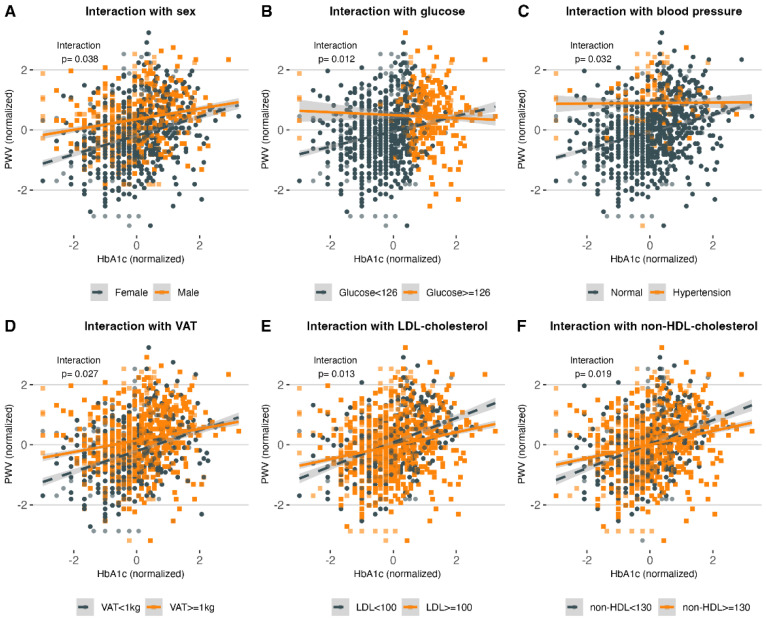
(**A**–**F**) show the interactions of sex, FGP, blood pressure, VAT, LDL-C, and non-HDL-C with HbA1c and PWV, respectively. The linear relationship between HbA1c and PWV is modified by sex, FGP ≥ 126 mg/dL, hypertension (SBP ≥ 140 mg/dL or DBP ≥ 90 mg/dL), visceral obesity (VAT ≥ 1 kg), LDL-C ≥ 100 mg/dL, and non-HDL-C ≥ 130 mg/dL. Abbreviations: HbA1c: Glycated hemoglobin. PWV: Pulse-wave velocity. FGP: fasting glucose plasma. VAT: Visceral adipose tissue. DBP: Diastolic blood pressure. SBP: Systolic blood pressure. HDL-C: High-density lipoprotein cholesterol. LDL-C: Low-density lipoprotein cholesterol.

**Table 1 ijerph-19-11017-t001:** Population demographic characteristics, laboratory data, and pulse-waveform analysis measurements in Hispanic subjects.

Variables	Overall *n* = 840	Without T2D or Prediabetes *n* = 161	Prediabetes *n* = 259	T2D *n* = 420	*p*-Value
Age, (years)	51 (41–59)	44 (32–54)	51 (41–57)	53.5 (43–62)	<0.001
Female sex, *n* (%)	544 (64.8)	110 (68.3)	182 (70.3)	252 (60)	0.014
HbA1c, (%)	6.1 (5.7–7.6)	5.4 (5.1–5.6)	5.9 (5.7–6.1)	7.5 (6.3–10)	<0.001
FPG, (mg/dL)	102.5 (93–127)	92 (86–96)	97 (91–104)	127 (104–194.75)	<0.001
TG, (mg/dL)	161 (106.8–232.3)	116 (78–165)	145 (98.5–206.5)	190 (133.5–258.8)	<0.001
TC, (mg/dL)	191 (165–223)	186 (161–224)	189 (167.5–217.5)	195 (165–226)	0.556
HDL-C, (mg/dL)	42 (36–51)	46 (39–54)	43 (36–52)	42 (35–49)	<0.001
LDL-C, (mg/dL)	117.7 (96.7–140.5)	113.8 (95.7–140.1)	119.2 (100.3–139)	117.5 (95.4–142.2)	0.715
non-HDL-C, (mg/dL)	148 (123–176)	145 (119–168)	145 (124.5–172.5)	151 (123.8–181.3)	0.057
RM, (mg/dL)	27.5 (20.7–36.8)	22.9 (16.9–29.6)	25.4 (20–34.1)	30.9 (23.6–39.7)	<0.001
Creatinine, (mg/dL)	0.7 (0.6–0.9)	0.7 (0.6–0.9)	0.7 (0.6–0.8)	0.7 (0.6–0.9)	0.093
BMI, (kg/m^2^)	28.9 (26.1–32.3)	27.8 (24.4–31.7)	29.5 (27.1–32.5)	28.9 (25.9–32.4)	<0.001
VFMI, (kg/m^2^)	0.5 (0.3–0.6)	0.3 (0.2–0.5)	0.5 (0.3–0.6)	0.5 (0.4–0.7)	<0.001
METS-IR	46.9 (40.8–53.5)	41.6 (35.4–48.7)	47.0 (40.7–52.6)	48.6 (42.7–55.2)	<0.001
SBP, (mmHg)	112 (104–122)	110 (101–116)	112 (103–120)	114 (106–124.5)	<0.001
DBP, (mmHg)	73 (67–80)	71 (65–76)	74 (67–80)	74 (68–81)	<0.001
PWV, (m/s)	6.4 (5.6–7.3)	5.9 (5.3–6.6)	6.1 (5.5–6.8)	6.8 (6–7.8)	<0.001
AIX75, (%)	34 (25–43)	31 (21–40)	36 (27–45)	34 (25–44)	0.002
Buckberg SEVR	148 (132–165)	151 (139–170)	153 (136.25–172)	143 (128–158)	<0.001

Values for continuous variables are given as medians and IQR; categorical variables are expressed as percentages (%). Abbreviations: T2D: Type 2 diabetes. HbA1c: Glycated hemoglobin A. FPG: fasting plasma glucose. TG: Triglycerides. TC: Total cholesterol. HDL-C: High-density lipoprotein cholesterol. LDL-C: Low-density lipoprotein cholesterol. RM: remnant cholesterol. BMI: Body mass index. VFMI: Visceral fat-mass index. METS-IR: Metabolic Score for Insulin Resistance. SBP: Systolic blood pressure. DBP: Diastolic blood pressure. PWV: Pulse-wave velocity. AIX75: Heart-rate-corrected augmentation index at 75 bpm. SEVR: Subendocardial viability ratio. The differences across groups were tested by a one-way analysis of variance.

**Table 2 ijerph-19-11017-t002:** Associations between HbA1c and pulse-waveform analysis measurements by multiple regression analysis.

Outcome	β-Coefficient	95% CI	Std Error	*p*-Value
PWV				
Model 1	0.179	0.107 to 0.252	0.037	<0.001
Model 2	0.090	0.034 to 0.146	0.028	0.002
Model 3	0.077	0.019 to 0.134	0.029	0.009
AIX75				
Model 1	0.865	−1.299 to 3.03	1.102	0.433
Model 2	−0.449	−2.305 to 1.408	0.946	0.635
Model 3	−0.785	−2.696 to 1.127	0.974	0.421
Buckberg SEVR				
Model 1	−0.103	−0.179 to −0.028	0.038	0.007
Model 2	−0.035	−0.093 to 0.023	0.029	0.234
Model 3	−0.036	−0.096 to 0.025	0.031	0.245

Model 1, with no adjustment. Model 2, adjusted by age, sex, HR, and MAP. Model 3, adjusted by METS-IR, VFMI, and LDL-cholesterol. Abbreviations: HbA1c: Glycated hemoglobin A. PWV: Pulse-wave velocity. AIX75: Heart-rate-corrected augmentation index at 75 bpm. SEVR: Subendocardial viability ratio. HR: Heart rate. MAP: Mean arterial blood pressure. METS-IR: Metabolic score for insulin resistance. VFMI: Visceral fat-mass index.

**Table 3 ijerph-19-11017-t003:** Mediation analyses of insulin resistance on the association between HbA1c and vascular health parameters (PWV and MAP).

Outcome	ACME	ADE	Total Effect	Proportion Mediated	*p*-Value
PWV	0.016 (0.004 to 0.031)	0.07 (0.009 to 0.127)	0.086 (0.028 to 0.14)	0.179 (0.04 to 0.66)	0.012
MAP	0.049 (0.03 to 0.072)	−0.027 (−0.094 to 0.036)	0.023 (−0.041 to 0.085)	1.114 (−16.477 to 16.505)	0.49

In all mediation models, the HbA1c and METS-IR were considered as effector and mediator, respectively, and were adjusted by age, sex, and LDL-cholesterol. In addition, adjustment by PWV and MAP was implemented where appropriate. Abbreviations: HbA1c: Glycated hemoglobin A. PWV: Pulse-wave velocity. MAP: Mean arterial blood pressure. ACME: Average Causal Mediation Effect. ADE: Average Direct Effect. METS-IR: Metabolic Score for Insulin Resistance.

## Data Availability

Not applicable.

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
