# Peer review of "Arterial Stiffness and HbA1c: Association Mediated by Insulin Resistance in Hispanic Adults"

_ijerph, 2022, doi:10.3390/ijerph191711017_

Round 1

Reviewer 1 Report

I read with great interest the manuscript entitled Arterial stiffness and HbA1c “association mediated by insulin resistance in obese subjects”. The authors aimed to investigate the association between arterial stiffness and HbA1c using a cross-sectional study in obese subjects. They found with a multivariate regression analysis that HbA1c is associated with PWV even after adjusting for several confounding variables.

Although I found the work interesting, I have some concern regarding its novelty. It’s well known that diabetes is a major determinant of vascular stiffness and the association of PWV and HbA1c has been demonstrated in several cohorts including subjects with hypertension and CKD. Even in young subjects with T1DM PWV is increased compared to controls (Christoforidis A et al, Diabetes Metab Res Rev, 2022). The authors should therefore clearly highlight the novelty of their findings.

The other major aspect which the authors might consideration is the role of confounders not considered in their statistical models. Medications for T2DM were not listed and those influence metabolic profile and potentially cardiovascular variables. Although blood pressure values were similar between the groups, cardiovascular medications might differ. Those should be included in the analysis. VFMI is significantly different among participants but was considered in the models.  

In table 1 and other part of the manuscripts, the authors compare subjects with pre-diabetes, diabetes and “normal subjects”. However, in the methodology it seems that patients with obesity and dyslipidaemia were recruited.  Can the authors clarify this point?

If only subjects with obesity were recruited, why many participants had a BMI < 30 kg/m2? For the same reasons, the title of the manuscript is misleading since average BMI in the population is 28.9. 

Overall the manuscript is difficult to read and would benefit from a more specific introduction and discussion to clarify the aim of the analysis and the novel findings. 

Reviewer 2 Report

Reviewer comments and suggestions

The study investigated the association between arterial stiffness and HbA1c and explain this effect was due to insulin resistance. For this purpose, the author's used cross-sectional study, arterial stiffness (pulse wave velocity; PWV), HbA1c and insulin resistance (METS-IR) were determined in obese subjects. The authors have used various parameters such as sex, age, glucose, lipid profile, etc along with adipose tissue (fat mass and visceral fat mass) to be considered as potential confounding variables.

The multiple regression study suggested that HbA1c is associated with PWV and importantly insulin resistance mediated 17.9% of the effect of HbA1c over PWV. Finally, the study concluded that HbA1c could be a potential resource for predicting arterial stiffness because of insulin resistance. 

Overall the manuscript is good for publication. However, few concerns are below to be incorporated in the revised version of the manuscript. 

  1. Line 36-37 Need more references here
  2. Line 43-44 there might be lacking relationship to discuss the hypothesis of the study. Could you please explore it in a better way?
  3. Line 59 probably there would be a number, kindly mention that
  4. Line 80 Why this parameter was important, could the authors explain here
  5. Line 92 Could you elaborate the reference here for calculating IR
  6. In the material and method section It would be nice if the author proposed a ray diagram/ flow chart of the study population
  7. Line 201-202 so these AIX75, SEVR: are the parameters of PWV?
  8. Line 215 How the authors have calculated the parameter ACME: Average Causal Mediation Effect. ADE: Average Direct Effect.
  9. Line 241, T2D; inversely Buckberg SEVR decreases as subjects progress into the disease. What does it mean?
  10. Line 248 is whether there were any previous studies for the sex-associated effect
  11. Line 262-263 what the authors want to state here, please explore with the reference here.
  12. If possible the authors can update more recent papers on PWV, glycosylated hemoglobin, and insulin resistance also few journal's style were wrong, please update it based on MDPI JOURNALS
